# Prognostic significance of index (LANR) composed of preoperative lymphocytes, albumin, and neutrophils in patients with stage IB-IIA cervical cancer

**Shan Wang**[1☯], **Yuan Wang**[1☯], **Jiaru Zhuang**[1], **Yibo Wu**[1], **Weifeng Shi**[2]*, **Lei Wang**[2]*

**1** Obstetrics, Gynecology and Reproduction Research, Affiliated Hospital of Jiangnan University, Wuxi, Jiangsu, P.R. China, **2** Department of General surgery, Affiliated Hospital of Jiangnan University, Wuxi, Jiangsu, P.R. China

☯ These authors contributed equally to this work.
* georgia914@sina.com (LW); 576095187@qq.com (WS)

## Abstract

**Data Availability Statement:** All relevant data are within the paper and its Supporting Information files.

### Background

The purpose of this study was to investigate the role of preoperative lymphocytes, albumin, neutrophils, and LANR in the prognosis of patients with stage IB-IIA cervical cancer (CC).

### Methods

We made a retrospective analysis of the clinical information and related materials of 202 patients with stage IB-IIA primary cervical cancer who had undergone a radical hysterectomy in the Department of Gynecology at the Affiliated Hospital of Jiangnan University between January 2017 and December 2018. The definition of LANR was as follows: LANR, lymphocyte × albumin / neutrophil. The receiver operating characteristic curve (ROC) was generated to determine the best cut-off values for these parameters, as well as the sensitivity and specificity of LANR in predicting recurrence and survival. The Kaplan–Meier method was employed to draw survival curves in our survival analysis. Univariate analysis, multifactorial analysis, and subgroup analysis were used to evaluate the prognostic significance of LANR in overall and progression-free survival.

### Results

The median follow-up time of the study was 55 months. In overall survival, the area under the curve for LANR was 0.704 (95% CI: 0.590–0.818, *p<0.05*). And in progression-free survival, the area under the curve for LANR was 0.745 (95% CI: 0.662–0.828, *p<0.05*). Univariate and multivariate analyses showed that the value of LANR was associated with both overall survival and progression-free survival (*p< 0.05*). Kaplan-Meier analysis demonstrated that OS (*p< 0.001*) and PFS (*p< 0.001*) in patients with high LANR levels were significantly higher than those with low LANR levels.

**Funding:** This research was supported by a grant from the National Natural Science Fund (81701511) and grants from the Top Talent Support Program for Young and Middle-Aged People of Wuxi Health Committee (BJ2020047). The funders had no role in study design, data collection and analysis, decision to publish, or preparation of the manuscript.

**Competing interests:** The authors have declared that no competing interests exist.

## Conclusions

Our findings suggested that LANR might serve as a clinically reliable and effective independent prognostic indicator in patients with stage IB-IIA cervical cancer.

## Introduction

Cervical cancer (CC) is the 4[th] most frequent malignancy in women worldwide and the leading malignancy of the female reproductive system [1]. It poses a serious threat to the health status of women. According to the 2020 World Health Statistics of the World Health Organization (WHO), cervical cancer accounts for 7.7% of all fatalities from malignant tumors in women, with an estimated 604,000 new cases and 342,000 deaths per year worldwide [2]. Cervical cancer incidence and mortality rates vary significantly from country (or region) to country (or region) and are closely related to the level of socio-economic development, with more than 80 percent of new cases and deaths from cervical cancer occurring in low- and middle-income countries or regions [3]. The incidence and mortality rates of cervical cancer increase as women age, with women aged 35–64 years being the main incidence group, and the incidence peaks at the age of 50–54 years [3]. In China, the incidence and mortality of cervical cancer are lower than the world average, but due to the large population base, the number of new cases and deaths of cervical cancer accounted for 18.2% and 17.3% of the global total cases, respectively [4].

Patients with early-stage cervical cancer (International Federation of Gynecology and Obstetrics [FIGO] stage IB-IIA) are typically treated with a radical hysterectomy combined with post-surgical chemoradiotherapy [5]. However, the fact remains that the cure rate after surgery is still unsatisfactory, with about 10–30% of cervical cancer patients still dying from recurrence or progression of the disease [6]. Many studies have shown that relying solely on FIGO staging to predict the prognosis of patients is not accurate in clinical practice [7]. Therefore, there is an urgent need to find out a non-invasive and easily accessible biomarker for proper prognostic assessment to reduce the recurrence of cervical cancer and improve patient survival.

Clinically, it is generally believed that human infection with the HPV virus is the main pathogenic factor of cervical cancer, especially human papillomavirus HPV16 and 18 types of infection, which cervical cancer is more closely related to [8]. Numerous pertinent research conducted recently discovered that the tumor microenvironment, inflammatory cell infiltration, and immunosuppression play a crucial role in the occurrence and progression of cervical cancer [9]. Many recent researches on tumor prognosis have shown that the patient's inflammatory response [10] and nutritional status [11] are associated with the prognosis of cancer. Among peripheral blood cells, lymphocytes, monocytes, and neutrophils are the most commonly used indicators of inflammation in response to the tumor microenvironment [12]. In addition, levels of C-reactive protein (CRP) and albumin (ALB), among circulating inflammatory proteins closely related to systemic nutritional status, are valuable prognostic indicators for many malignancies, including cervical cancer [13]. Furthermore, the integration of these laboratory indicators, such as neutrophil/lymphocyte (NLR) [14], systemic inflammatory response index (SIRI) [12], C- reactive protein/albumin (CAR) [13], and prognostic nutritional index (PNI) [15] are effective in improving the accuracy of predicting the prognosis of cervical cancer. However, the prognostic value of these indicators, which focuses only on the inflammatory response or nutritional status, is limited [16]. We need to evaluate the inflammatory and nutritional status of patients to obtain a more accurate prognosis comprehensively.

The LANR is a new comprehensive scoring index based on inflammatory response and nutritional status, constructed from a combination of three indicators: lymphocytes, albumin, and neutrophils. Previous studies have shown that the LANR index is associated with the prognosis of patients with colorectal cancer (CRC) [17]. However, there have been no studies to date on the predictive importance of LANR in cervical cancer. Therefore, the aim of this study was to assess the prognostic value of LANR in patients with stage IB-IIA cervical cancer, to predict their survival, and to offer personalized treatment to patients.

## Materials and methods

### Study design and patients

We retrospectively collected relevant data on patients with cervical cancer who underwent initial treatment with a radical hysterectomy at the Affiliated Hospital of Jiangnan University between January 2017 and December 2018. Data collection and data access dates for research purposes began in July 2022. The cancer staging for each patient was based on the 2009 FIGO staging system [18] determined. The inclusion criteria were as follows: (1) patients with pathologically diagnosed cervical cancer with FIGO stage IB-IIA; (2) all underwent radical hysterectomy for the first time; (3) with complete clinicopathological and laboratory data; (4) with five-year follow-up information. The exclusion criteria were as follows: (1) with a history of malignancy or combined with other primary tumors; (2) without complete clinical and follow-up data; (3) with hematological disorders or autoimmune diseases; (4) who underwent neoadjuvant therapy before surgery. This research was approved by the Medical Ethics Committee of the Affiliated Hospital of Jiangnan University (JNMS01202200172). Because all data were anonymized and aggregated, the requirement for informed consent was waived by the ethics committee. The study was executed following the principles of the Declaration of Helsinki.

### Treatment

The diagnosis and treatment of CC patients were based on the latest international advances in cervical cancer at the time. Radical hysterectomy and bilateral pelvic lymphadenectomy were performed on all patients who were recruited. Postoperative adjuvant chemoradiotherapy was determined according to the pathological report. Following the Sedlis criteria [19], patients with single or multiple intermediate risk factors (such as lympho-vascular space invasion, deep stromal invasion, and a large primary tumor) and patients with any combination of high-risk factors (such as positive pelvic lymph nodes, positive surgical margins, and parametrial invasion) were advised to receive adjuvant radiotherapy (ART) or concurrent chemoradiotherapy. The recurrence of the tumor was confirmed by two experienced gynecological oncologists who were blinded to this study design.

### Follow-up

Through the hospital's electronic medical record system, we gathered clinical information about cervical cancer patients, including pathology data and preoperative blood biochemical indicators, such as age, height, weight, tumor grade, FIGO stage, tumor size, lymph node metastasis, parametrial invasion, lympho-vascular space invasion, depth of invasion, radiotherapy, chemotherapy, surgical type, ALB, lymphocytes, and neutrophils. Follow-up information was obtained from outpatient reviews, inpatient reviews, and telephone and e-mail interviews. The first post-surgical follow-up for cervical cancer was approximately two months after surgery, and patients were required to review pelvic CT, MRI, and blood tests including tumor markers, and subsequently every three months. If there was no recurrence within two

years, it could be repeated every six months until five years. The follow-up period was defined as the date of treatment initiation to the date of final confirmation of patient survival or death. The study was followed up until November 2022. Overall survival (OS) was designated as the primary outcome and progression-free survival (PFS) as the secondary outcome. OS was defined as the period from the date of surgery till death from any cause or the end of follow-up, and PFS was defined as the period from the date of surgery to the first sign of disease progression or the end of follow-up. The following formula was used to calculate the laboratory data: LANR, lymphocyte × albumin / neutrophil.

## Statistical analyses

Analysis was performed using the statistical software SPSS 26.0. Categorical variables were expressed as percentages, while continuous variables were expressed as median (interquartile range) or mean ± standard deviation. The differences between categorical variables were calculated using either the chi-square test or Fisher's exact test. For continuous variables, the Student's t-test or Wilcoxon test was employed to compare differences between the two groups. The area under the curve and the Jorden index was calculated using the receiver operating characteristic curve (ROC). OS and PFS analyses were performed using Kaplan-Meier survival curves and log-rank tests. Variables with $p<0.05$ in univariate analysis were included in Cox proportional hazards regression models for multivariate analysis to identify factors independently linked with cumulative survival. Forest plots were used to display the findings of subgroup studies to determine the prognostic correlations between patients with various features and LANR. $P$-values $< 0.05$ were considered statistically significant.

## Results

### Patient characteristics

This study included 202 patients with stage IB-IIA cervical cancer who underwent radical hysterectomy plus pelvic lymph node dissection. The clinical characteristics of 202 patients are shown in Table 1. All were female. The median age of the patients was 49 years (range, 25–73 years). All tumors were pathologically staged after surgery, with 144 patients (71.29%) in stage IB and 58 (28.71%) in stage IIA. A total of 44 (21.78%) patients deteriorated and 26 patients (12.87%) died. The study's median follow-up period was 55 months.

### Prognostic value of LANR for overall survival

In terms of overall survival, the area under the ROC curve (AUC) and the optimal threshold values of albumin, lymphocytes, neutrophils, and LANR for OS are shown in Table 2. Based on the ROC curve, we discovered that the area under the curve for LANR was the best, with an AUC of 0.704 and an optimal cut-off value of 19.42 (sensitivity: 65.3%, specificity: 76.9%, $p < 0.001$; see Fig 1). According to the cut-off value, enrolled patients were divided into high-level (LANR $\geq$ 19.42, n = 81, 40.1%) and low-level (LANR $<$ 19.42, n = 121, 59.9%) groups (see Table 3), with Kaplan-Meier survival curves showing longer overall survival for patients with higher LANR (see Fig 2). Univariate analysis revealed that FIGO stage, tumor size, lymph node metastasis, parametrial invasion, depth of invasion, radiotherapy, chemotherapy, albumin, lymphocytes, neutrophils, and LANR were significantly associated with OS ($p<0.05$; see Table 2). Multivariate analysis revealed that high levels of albumin, lymphocytes, neutrophils, and LANR had 0.560 (95% CI: 0.225–1.398), 1.146 (95% CI: 0.414–3.171), 1.800 (95% CI: 0.680–4.768) and 0.245 (95% CI: 0.067–0.894)-fold risk of death, respectively (see Table 2). There was a significant association between LANR and OS in patients of different ages (<50

**Table 1. Clinicopathological Characteristics of 202 Patients with stage IB-IIA Cervice Cancer.**

| | | Disease progression | | P-value | Death | | P-value |
|---|---|---|---|---|---|---|---|
| | | **Without** | **With** | | **Without** | **With** | |
| | | **n = 158(%)** | **n = 44(%)** | | **n = 176(%)** | **n = 26(%)** | |
| Age (years) | | 48(41–55) | 51.5(45.3–57.8) | 0.101 | 48(42.3–55) | 51.5(46.5–58.3) | 0.050 |
| BMI (kg/m$^2$) | | 22.95±3.08 | 23.55±2.48 | 0.352 | 22.94±3.03 | 23.98±2.35 | 0.334 |
| Pathological type | SCC | 148(93.67) | 40(90.91) | 0.510 | 164(93.18) | 24(92.31) | 0.870 |
| | Non-SCC | 10(6.33) | 4(9.09) | | 12(6.82) | 2(7.69) | |
| Tumor grade | G1 | 19(12.03) | 2(4.55) | 0.042 | 19(10.80) | 2(7.69) | 0.146 |
| | G2 | 71(44.94) | 14(31.82) | | 78(44.32) | 7(26.92) | |
| | G3 | 68(43.04) | 28(63.64) | | 79(44.89) | 17(65.38) | |
| FIGO stage | IB1 | 79(50.00) | 8(18.18) | 0.002 | 83(47.16) | 4(15.38) | 0.012 |
| | IB2 | 41(25.95) | 16(36.36) | | 48(27.27) | 9(34.62) | |
| | IIA1 | 21(13.29) | 10(22.73) | | 25(14.20) | 6(23.08) | |
| | IIA2 | 17(10.76) | 10(22.73) | | 20(11.36) | 7(26.92) | |
| Tumor size | <4cm | 100(63.29) | 18(40.91) | 0.008 | 108(61.36) | 10(38.46) | 0.027 |
| | ≥4cm | 58(36.71) | 26(59.09) | | 68(38.64) | 16(61.54) | |
| LNM | No | 141(89.24) | 18(40.91) | <0.001 | 150(85.23) | 9(34.62) | <0.001 |
| | Yes | 17(10.76) | 26(59.09) | | 26(14.77) | 17(65.38) | |
| Parametrial invasion | No | 158(100.00) | 38(86.36) | <0.001 | 174(98.86) | 22(84.62) | 0.003 |
| | Yes | 0(0.00) | 6(13.64) | | 2(1.14) | 4(15.38) | |
| LVSI | No | 84(53.16) | 16(36.36) | 0.049 | 91(51.70) | 9(34.62) | 0.104 |
| | Yes | 74(46.85) | 28(63.64) | | 85(48.30) | 17(65.38) | |
| Invasion depth | <2/3 | 92(58.23) | 13(29.55) | 0.001 | 100(56.82) | 5(19.23) | <0.001 |
| | ≥2/3 | 66(41.77) | 31(70.45) | | 70(39.77) | 21(80.77) | |
| Radiotherapy | No | 126(79.75) | 16(36.36) | <0.001 | 134(76.14) | 8(30.77) | <0.001 |
| | Yes | 32(20.25) | 28(63.64) | | 42(23.86) | 18(69.23) | |
| Chemotherapy | No | 74(46.84) | 10(22.73) | 0.004 | 79(44.89) | 5(19.23) | 0.013 |
| | Yes | 84(53.16) | 34(77.27) | | 97(55.11) | 21(80.77) | |
| Surgery | Open | 103(65.19) | 29(65.91) | 0.929 | 113(6 4.20) | 19(73.08) | 0.375 |
| | Laparoscopic | 55(34.81) | 15(34.09) | | 63(35.80) | 7(26.92) | |
| ALB(G/L) | | 41.55(39.8–43.33) | 40.15(38.48–42.6) | 0.029 | 41.5(39.7–43.28) | 40.45(38.7–42.75) | 0.287 |
| Lym(10$^9$/L) | | 1.8(1.5–2.2) | 1.6(1.3–1.98) | 0.036 | 1.8(1.5–2.2) | 1.55(1.28–2.05) | 0.138 |
| Neu(10$^9$/L) | | 3.4(2.8–4.0) | 4(3.4–5.18) | 0.021 | 3.5(2.8–4.08) | 4.1(3.35–5.33) | 0.066 |

Abbreviations: BMI, body mass index; SCC, squamous cell carcinoma; FIGO, International Federation of Gynecology and Obstetrics; LNM, lymphatic node metastasis; LVSI, lympho-vascular space invasion; ALB, albumin; Lym, lymphocyte; Neu, neutrophil.

P-values were calculated by the Student's t-test or Wilcoxon test for continuous variables, and the Chi-square test or Fisher's exact test for categorical variables, respectively.

years), BMI(<23kg/m$^2$), pathological type (SCC), FIGO stage (IB), and tumor grade(G3) (see Fig 3). In patients with stage IB-IIA cervical cancer, we discovered that LANR was a reliable predictor of overall survival in combination with the area under the ROC curve and multivariate Cox regression analysis.

## Prognostic value of LANR for progression-free survival

In terms of progression-free survival, the area under the ROC curve (AUC) and the optimal threshold values of albumin, lymphocytes, neutrophils, and LANR for PFS are shown in Table 4. Based on the ROC curve, we discovered that the area under the curve for LANR was

**Table 2. Risk factors for OS in CC patients with stage IB-IIA by univariate and multiple Cox regression analysis.**

| | AUC | Cut-Point | Univariate | | Multivariate | |
|---|---|---|---|---|---|---|
| | | | HR(95%CI) | P-value | HR(95%CI) | P-value |
| ALB(G/L) | 0.591 | 40.05 | 0.442(0.205–0.953) | 0.037 | 0.560(0.225–1.398) | 0.214 |
| Lym($10^9$/L) | 0.635 | 1.65 | 0.417(0.192–0.909) | 0.028 | 1.146(0.414–3.171) | 0.793 |
| Neu($10^9$/L) | 0.668 | 3.95 | 3.946(1.790–8.699) | 0.001 | 1.800(0.680–4.768) | 0.237 |
| LANR | 0.704 | 19.42 | 0.180(0.072–0.449) | <0.001 | 0.245(0.067–0.894) | 0.033 |
| FIGO stage | | | 2.563(1.188–5.528) | 0.016 | 1.728(0.715–4.174) | 0.224 |
| Tumor size | | | 2.395(1.087–5.279) | 0.030 | 0.727(0.267–1.978) | 0.533 |
| LNM | | | 8.368(3.726–18.796) | <0.001 | 7.043(1.660–29.874) | 0.008 |
| Parametrial invasion | | | 11.180(3.815–32.762) | <0.001 | 3.315(0.898–12.231) | 0.072 |
| Depth of invasion | | | 5.005(1.887–13.276) | 0.001 | 2.545(0.799–8.107) | 0.114 |
| Radiotherapy | | | 5.972(2.595–13.741) | <0.001 | 0.741(0.147–3.740) | 0.716 |
| Chemotherapy | | | 3.188(1.202–8.456) | 0.020 | 1.504(0.388–5.836) | 0.555 |

Abbreviations: HR, Hazard Ratio; CI, Confidence Interval; AUC, Area under the ROC Curve; Lym, lymphocyte; ALB, albumin; Neu, neutrophils; LANR, Lym*Alb/Neu; FIGO, International Federation of Gynecology and Obstetrics; LNM lymphatic node metastasis

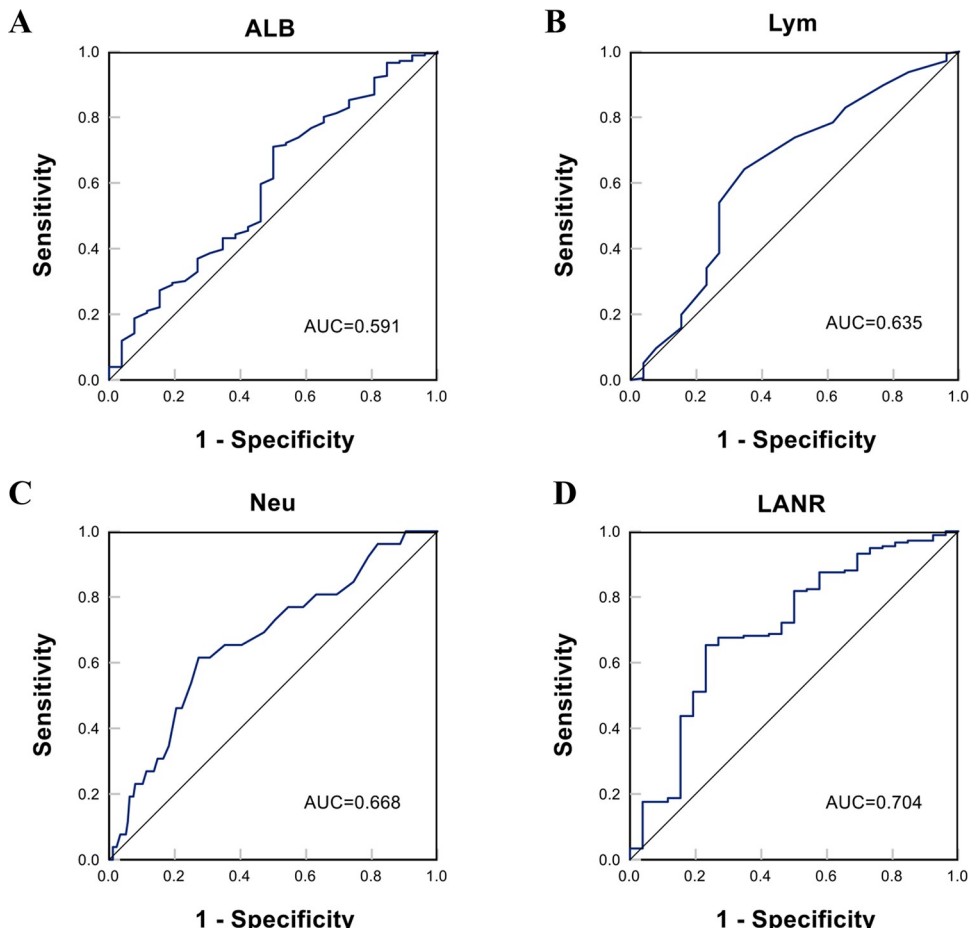

**Fig 1. The ROC curve for overall survival of Alb, Lym, Neu, and LANR.** A: Alb for OS. B: Lym for OS. C: Neu for OS. D: LANR for OS.

**Table 3. Relationship between clinicopathological characteristics of overall survival in patients with stage IB-IIA cervical cancer and LANR.**

| | | LANR value | | P-value |
|---|---|---|---|---|
| | | Low | High | |
| | | (n = 81) (%) | (n = 121) (%) | |
| Age (years) | | 49(44–55) | 50(41.5–56.5) | 0.551 |
| BMI (kg/m$^2$) | | 23.10±3.09 | 23.06±2.89 | 0.613 |
| Pathological type | SCC | 76 (93.83) | 112(92.56) | 0.729 |
| | Non-SCC | 5(6.17) | 9(7.44) | |
| Tumor grade | G1 | 9(11.11) | 12(9.92) | 0.935 |
| | G2 | 33(40.74) | 52(42.98) | |
| | G3 | 39(48.15) | 57(47.11) | |
| FIGO stage | IB1 | 24(29.63) | 63(53.07) | <0.001 |
| | IB2 | 31(38.27) | 26(21.49) | |
| | IIA1 | 9(11.11) | 22(18.18) | |
| | IIA2 | 17(20.99) | 10(8.26) | |
| Tumor size | <4cm | 33(40.74) | 85(70.25) | <0.001 |
| | ≥4cm | 48(59.26) | 36(29.75) | |
| LNM | No | 57(70.37) | 102(84.30) | 0.018 |
| | Yes | 24(29.63) | 19(15.70) | |
| Parametrial invasion | No | 77(95.06) | 119(98.35) | 0.221 |
| | Yes | 4(4.94) | 2 (1.65) | |
| LVSI | No | 45(55.56) | 55(45.45) | 0.159 |
| | Yes | 36(44.44) | 66(54.55) | |
| Invasion depth | <2/3 | 35(43.21) | 70(57.85) | 0.041 |
| | ≥2/3 | 46(56.79) | 51(42.15) | |
| Radiotherapy | No | 47(58.02) | 95(78.51) | 0.002 |
| | Yes | 34(41.98) | 26(21.49) | |
| Chemotherapy | No | 34(41.98) | 50(41.32) | 0.926 |
| | Yes | 47(58.02) | 71(58.68) | |
| Surgery | Open | 61(75.31) | 71(58.68) | 0.015 |
| | Laparoscopic | 20(24.69) | 50(41.32) | |

LANR, Lym
*Alb/Neu.

the best, with an AUC of 0.745 and an optimal cut-off value of 19.17 (sensitivity: 72.8%, specificity: 75.0%, $p < 0.001$; see Fig 4). According to the cut-off value, enrolled patients were divided into high-grade (LANR ≥ 19.17, n = 76, 37.6%) and low-grade (LANR < 19.17, n = 126, 62.4%) groups (see Table 5), with Kaplan- Meier survival curves showing longer progression-free survival in patients with high-grade LANR (see Fig 5). Univariate analysis revealed that tumor grade, FIGO stage, tumor size, lymph node metastasis, parametrial invasion, lympho-vascular space invasion, depth of invasion, radiotherapy, chemotherapy, albumin, lymphocytes, neutrophils, and LANR were all significantly associated with PFS ($p < 0.05$; see Table 4). Multivariate analysis revealed that high levels of albumin, lymphocytes, neutrophils, and LANR had 0.497 (95% CI: 0.240–1.031), 1.523 (95% CI: 0.650–3.566), 1.384 (95% CI: 0.649–2.952) and 0.180 (95% CI: 0.064–0.503)-fold risk of disease progression, respectively (see Table 4). There was a significant association between LANR and PFS in patients of different ages, BMI, pathological type (SCC), FIGO stage, and tumor grade(G3) (see Fig 6). In patients with stage IB-IIA cervical cancer, we discovered that LANR was a reliable predictor of

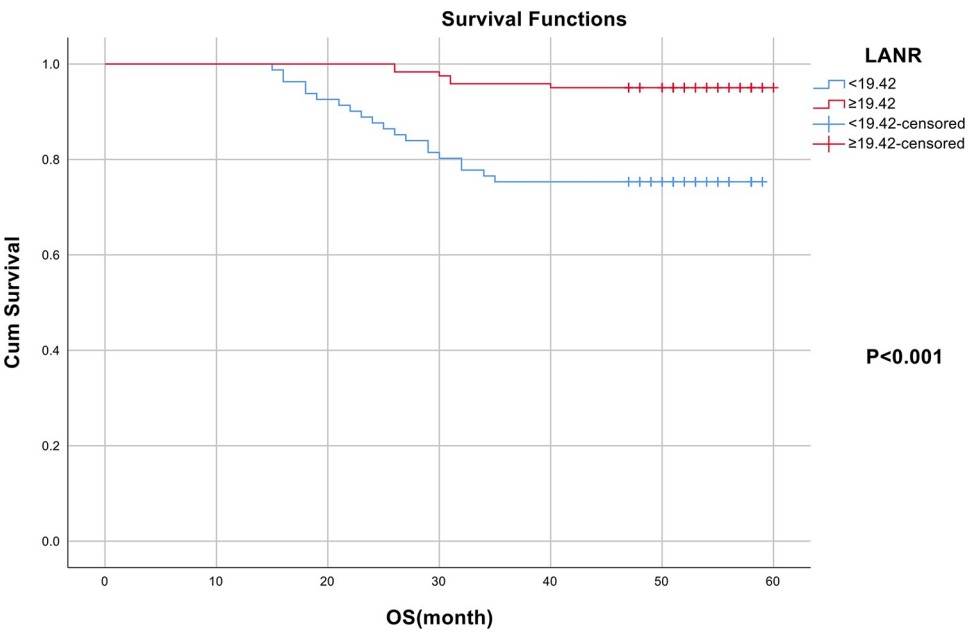

**Fig 2. The Kaplan–Meier curves for overall survival of cervical cancer patients based on LANR.**

five-year progression-free survival in combination with the area under the ROC curve and multivariate Cox regression analysis.

## Discussion

In this study, we investigated the predictive value of 13 clinicopathological variables (age, BMI, pathological type, tumor grade, FIGO stage, tumor size, lymph node metastasis, parametrial

| Subgroup | No.of Patients | Death | | Hazard Ratio (95%CI) | P for trend |
|---|---|---|---|---|---|
| Overall | 202 | 26 | | | |
| Age | | | | | |
| < 50yr | 102 | 10 | | 0.846 (0.770-0.930) | 0.001 |
| ≥50yr | 100 | 16 | | 0.951 (0.896-1.010) | 0.104 |
| BMI | | | | | |
| < 23kg/m² | 99 | 10 | | 0.880 (0.808-0.957) | 0.003 |
| ≥23kg/m² | 103 | 16 | | 0.926 (0.855-1.003) | 0.058 |
| Pathological type | | | | | |
| SCC | 188 | 24 | | 0.919 (0.870-0.970) | 0.002 |
| Non-SCC | 14 | 2 | | 0.904 (0.758-1.079) | 0.265 |
| FIGO stage | | | | | |
| IB1/IB2 | 144 | 13 | | 0.895 (0.830-0.965) | 0.004 |
| IIA1/IIA2 | 58 | 13 | | 0.945 (0.880-1.014) | 0.115 |
| Tumor grade | | | | | |
| G1 | 21 | 2 | | 0.971 (0.809-1.165) | 0.753 |
| G2 | 85 | 7 | | 0.997 (0.915-1.087) | 0.946 |
| G3 | 96 | 17 | | 0.890 (0.834-0.948) | < 0.001 |

**Fig 3. Forest plots of the associations of LANR with the overall survival of stage IB-IIA cervical cancer patients in different subgroups.**

**Table 4. Risk factors for PFS in CC patients with stage IB-IIA by univariate and multiple Cox regression analysis.**

|  | AUC | Cut-Point | Univariate | | Multivariate | |
|---|---|---|---|---|---|---|
|  |  |  | HR(95%CI) | P-value | HR(95%CI) | P-value |
| ALB(G/L) | 0.626 | 40.25 | 0.414(0.229–0.749) | 0.004 | 0.497(0.240–1.031) | 0.060 |
| Lym($10^9$/L) | 0.614 | 1.75 | 0.533(0.290–0.977) | 0.042 | 1.523(0.650–3.566) | 0.333 |
| Neu($10^9$/L) | 0.678 | 3.95 | 2.834(1.567–5.124) | 0.001 | 1.384(0.649–2.952) | 0.400 |
| LANR | 0.745 | 19.17 | 0.196(0.103–0.376) | <0.001 | 0.180(0.064–0.503) | 0.001 |
| Tumor grade |  |  | 2.119(1.146–3.918) | 0.017 | 0.926(0.415–2.067) | 0.852 |
| FIGO stage |  |  | 2.249(1.242–4.074) | 0.007 | 1.729 (0.900–3.324) | 0.100 |
| Tumor size |  |  | 2.228(1.221–4.066) | 0.009 | 0.811(0.369–1.783) | 0.602 |
| LNM |  |  | 7.548(4.127–13.806) | <0.001 | 9.547(2.963–30.764) | <0.001 |
| Parametrial invasion |  |  | 13.888(5.785–33.343) | <0.001 | 9.343(2.829–30.858) | <0.001 |
| LVSI |  |  | 0.523(0.283–0.967) | 0.039 | 0.912(0.359–2.317) | 0.846 |
| Depth of invasion |  |  | 3.055(1.598–5.842) | 0.001 | 1.668(0.753–3.695) | 0.207 |
| Radiotherapy |  |  | 5.211(2.816–9.642) | <0.001 | 0.495(0.140–1.750) | 0.275 |
| Chemotherapy |  |  | 2.788(1.377–5.644) | 0.004 | 1.769(0.637–4.917) | 0.274 |

Abbreviations: HR, Hazard Ratio; CI, Confidence Interval; AUC, Area under the ROC Curve; Lym, lymphocyte; ALB, albumin; Neu, neutrophils; LANR, Lym

*Alb/Neu; FIGO, International Federation of Gynecology and Obstetrics; LNM, lymphatic node metastasis; LVSI, lympho-vascular space invasion.

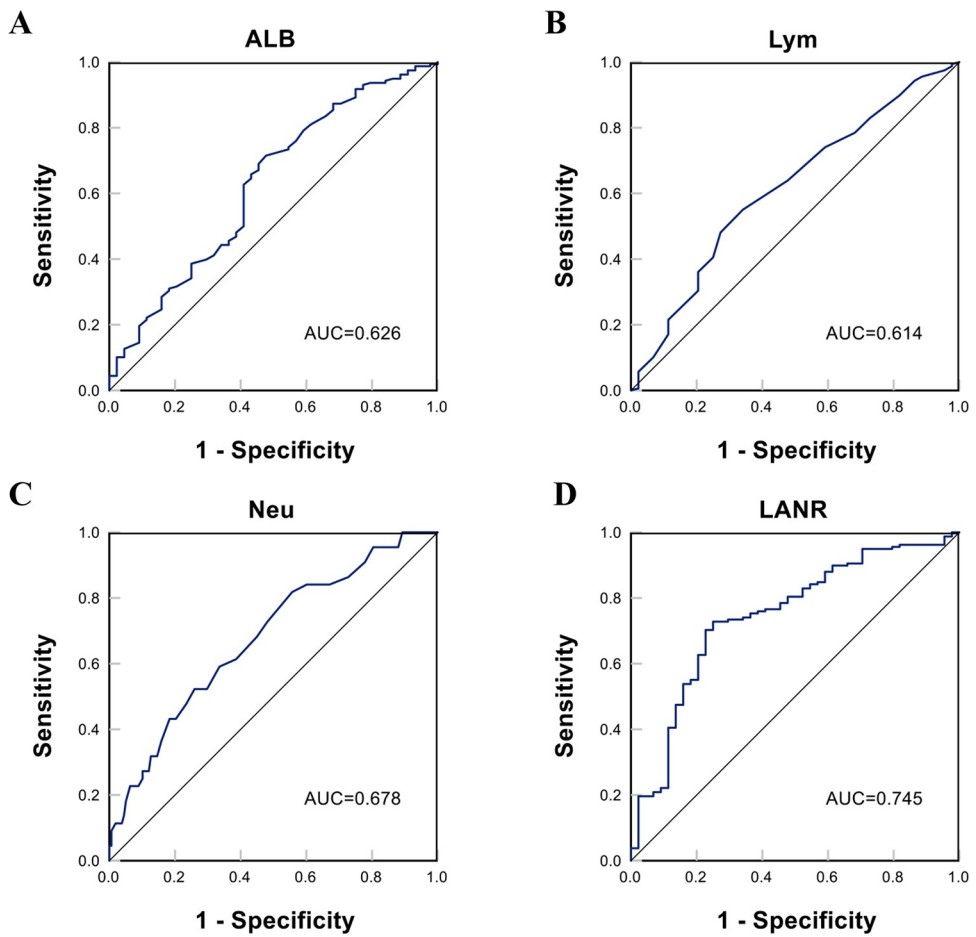

**Fig 4. The ROC curve for progression-free survival of Alb, Lym, Neu, and LANR.** A: Alb for PFS. B: Lym for PFS. C: Neu for PFS. D: LANR for PFS.

**Table 5. Relationship between clinicopathological characteristics of progression-free survival in patients with stage IB-IIA cervical cancer and LANR.**

| | | LANR value | | P-value |
| --- | --- | --- | --- | --- |
| | | Low | High | |
| | | (n = 76) (%) | (n = 126) (%) | |
| Age (years) | | 62(52–68) | 59(51–66) | 0.008 |
| BMI (kg/m$^2$) | | 23.10±3.09 | 23.06±2.89 | 0.613 |
| Pathological type | SCC | 71(93.42) | 117(92.86) | 0.878 |
| | Non-SCC | 5(6.58) | 9(7.14) | |
| Tumor grade | G1 | 8(10.53) | 13(10.32) | 0.665 |
| | G2 | 29(38.16) | 56(44.44) | |
| | G3 | 39(51.32) | 57(45.24) | |
| FIGO stage | IB1 | 23(30.26) | 64(50.79) | 0.001 |
| | IB2 | 29(38.16) | 28(22.22) | |
| | IIA1 | 8 (10.53) | 23(18.25) | |
| | IIA2 | 16(21.05) | 11(8.73) | |
| Tumor size | <4cm | 31(40.79) | 87(69.05) | <0.001 |
| | ≥4cm | 45(59.21) | 39(30.95) | |
| LNM | No | 53(69.74) | 106(84.13) | 0.016 |
| | Yes | 23(30.26) | 20(15.87) | |
| Parametrial invasion | No | 72(94.74) | 124(98.41) | 0.201 |
| | Yes | 4(5.26) | 2(1.59) | |
| LVSI | No | 42(55.26) | 58 (46.03) | 0.204 |
| | Yes | 34(44.74) | 68 (53.97) | |
| Invasion depth | <2/3 | 32(42.11) | 73(57.94) | 0.029 |
| | ≥2/3 | 44(57.89) | 53(42.06) | |
| Radiotherapy | No | 44(57.89) | 98(77.78) | 0.003 |
| | Yes | 32(42.11) | 28(22.22) | |
| Chemotherapy | No | 31(40.79) | 53(42.06) | 0.859 |
| | Yes | 45(59.21) | 73 (57.94) | |
| Surgery | Open | 45(59.21) | 75(59.52) | <0.001 |
| | Laparoscopic | 31(40.79) | 51(40.48) | |

LANR, Lym*Alb/Neu.

invasion, lympho-vascular space invasion, depth of myometrial invasion, radiotherapy, chemotherapy [20], and surgical type [21] and 3 systemic inflammatory parameters as well as LANR in stage IB-IIA cervical cancer. Among them, LNM, parametrial invasion, LVSI, and tumor size have been proven to be key prognostic factors for tumor recurrence and progression [22]. However, these indicators need to be confirmed by pathological examination after surgery, and it takes a certain amount of time. LANR, which consists of three important substances, namely lymphocytes, albumin, and neutrophils, is a reusable and inexpensive laboratory hematology indicator. With the help of this study, we demonstrated that LANR is a reliable predictive factor for patients with stage IB-IIA CC. Furthermore, according to our knowledge, this study is the first one that we are aware of to investigate the prognostic significance of LANR in cervical cancer.

The immune system plays an important role in the occurrence and progression of tumors. Although the immune system cannot suppress tumor production, the immune response within the established tumor microenvironment is an important factor in determining the prognosis of cancer. Nutrition is closely linked to the immune system. Impaired nutritional

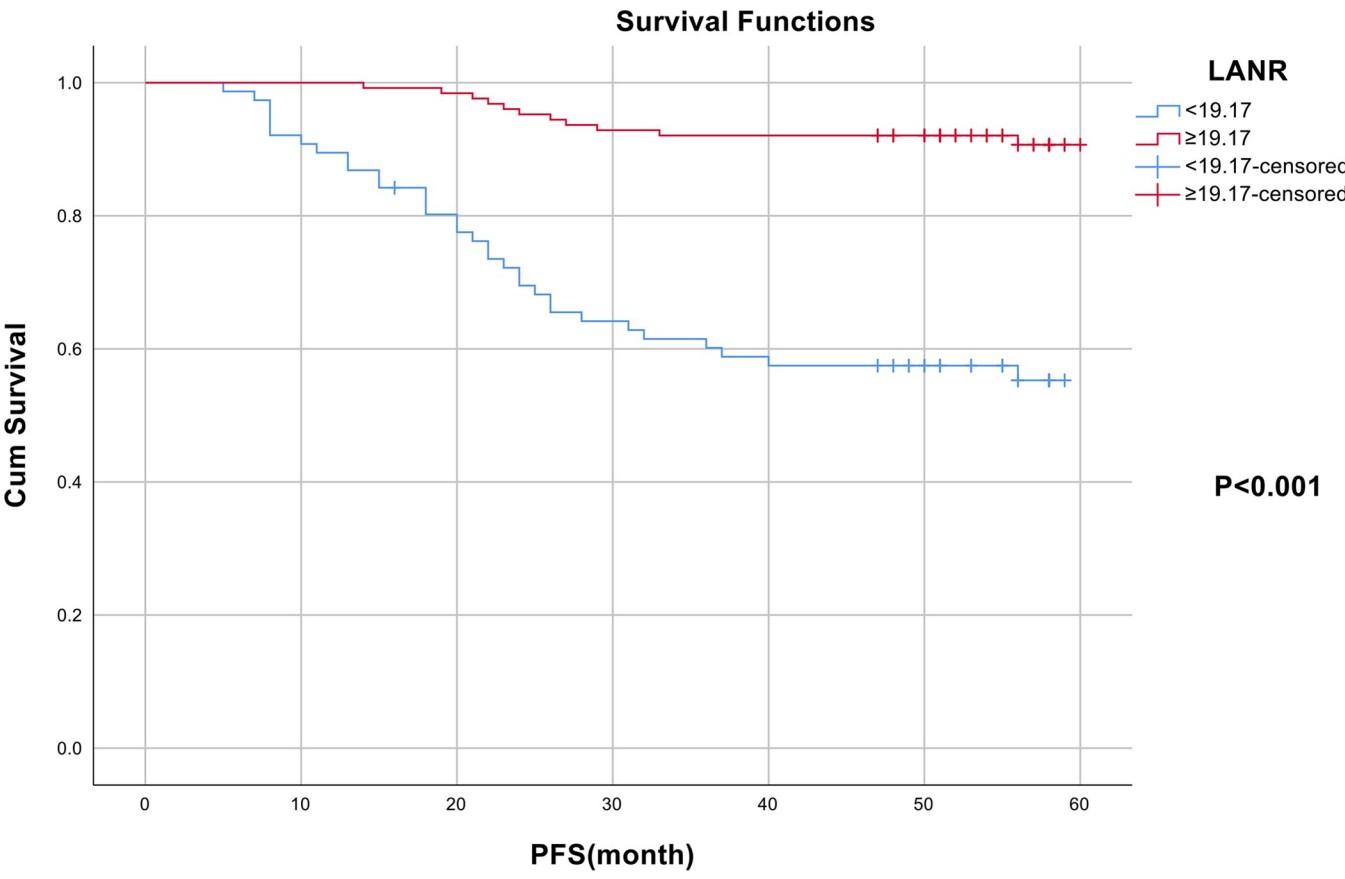

**Fig 5. The Kaplan–Meier curves for progression-free survival of cervical cancer patients based on LANR.**

status can suppress the anti-cancer immune response, leading to the acceleration of tumor progression. Serum albumin, the most important protein in human plasma, maintains body nutrition and osmolality and is the most commonly used indicator for clinical assessment of a patient's nutritional status [23]. It is the most commonly used clinical indicator to assess the nutritional status of patients. According to pertinent research, patients with low blood albumin levels had shorter overall survival and progression-free survival, which is related to the impact of nutritional impairment as seen by low serum albumin levels. Previous findings have confirmed that serum albumin can be an independent risk factor for the prognosis of several malignancies, including colon cancer [24], gastric cancer [25], small-cell lung cancer [26], and esophageal squamous cell carcinoma [27]. Therefore, the decrease in albumin is a predictor of poor outcomes in patients with tumors [28]. Lymphocytes, a fundamental component of the immune system, are dominated by T-cell- mediated cellular immunity [29]. Elevated levels of CD8$^+$, CD8$^+$/CD4$^+$, and FOXP3$^+$ (major transcription factors of the regulatory T cell lineage) in preoperative tumor-infiltrating T lymphocytes were found to be associated with negative lymph nodes, early disease, and radical hysterectomy [30]. T lymphocytes can not only effectively recognize tumor cells, but also be stimulated by tumor antigens to transform into sensitized lymphocytes and attack tumor lesions, thus generating antitumor activity, inhibiting angiogenesis, and improving tumor prognosis [31]. Neutrophils are the most abundant leukocytes in human circulation and have a pro-inflammatory effect [32]. Neutrophils in the tumor microenvironment also have similar functions to myeloid suppressor cells, inhibiting T cell proliferation, binding to tumor cells, and supporting tumor cell cycle progression [33]. High

| Subgroup | No.of Patients | Progression | | Hazard Ratio (95%CI) | P for trend |
|---|---|---|---|---|---|
| Overall | 1 | 44 | | | |
| Age | | | | | |
| < 50yr | 102 | 17 | | 0.858 (0.794-0.927) | < 0.001 |
| ≥50yr | 100 | 27 | | 0.926 (0.883-0.971) | 0.002 |
| BMI | | | | | |
| < 23kg/m² | 99 | 20 | | 0.873 (0.820-0.929) | < 0.001 |
| ≥23kg/m² | 103 | 24 | | 0.923 (0.865-0.985) | 0.015 |
| Pathological type | | | | | |
| SCC | 188 | 40 | | 0.903 (0.865-0.943) | < 0.001 |
| Non-SCC | 14 | 4 | | 0.918 (0.799-1.055) | 0.228 |
| FIGO stage | | | | | |
| IB1/IB2 | 144 | 24 | | 0.884 (0.836-0.936) | < 0.001 |
| IIA1/IIA2 | 58 | 20 | | 0.936 (0.883-0.993) | 0.029 |
| Tumor grade | | | | | |
| G1 | 21 | 2 | | 0.971 (0.809-1.165) | 0.753 |
| G2 | 85 | 14 | | 0.946 (0.883-1.014) | 0.117 |
| G3 | 96 | 28 | | 0.887 (0.842-0.934) | < 0.001 |

0.5   1.0   1.5   2.0   2.5

**Fig 6. Forest plots of the associations of LANR with the progression-free survival of stage IB-IIA cervical cancer patients in different subgroups.**

baseline neutrophil counts are a stable, independent factor in poor tumor prognosis, both in tumor tissue and in peripheral blood [34]. Notably, the role of neutrophils as a negative prognostic factor was not diminished by targeted therapy or gene therapy, but rather patients with low baseline levels had a better clinical prognosis on treatment [35]. In our study, ALB, Lym, and Neu were all associated with OS and PFS in CC in univariate analysis, however, they were not statistically significant in PFS and OS in a multifactorial analysis, and although they influenced PFS and OS, they were not independent prognostic factors. LANR was an indicator combining all three, lymphocytes, albumin, and neutrophils. Compared with these three indicators, LANR had a more significant prognostic effect. We found that this indicator was significantly associated with overall survival (95% CI: 0.590–0.818, $p < 0.05$) and progression-free survival (95% CI: 0.662–0.828, $p < 0.05$), and was an independent predictive factor for patients with stage IB-IIA cervical cancer.

This study has several limitations. Firstly, it was a single-center retrospective study, which makes it subject to selection bias and confounding factors. And the small sample size (202 cases) was not sufficient to fully reflect the prognosis of all cervical cancer patients, therefore, a large prospective validation study with multicentre data analysis is needed to further validate this result. Secondly, the new 2018 FIGO staging system [36] redefined lymph node metastasis for staging and refined stage IB, however, the patient data included in this study were mainly classified according to the 2009 FIGO staging system. Finally, the latest international ESGO guideline suggests not performing surgery for locally advanced cervical cancer (LACC). Simultaneous radiotherapy and chemotherapy should be preferred for patients with narrowly conceptualized locally advanced (stage IB3 and IIA2) cervical cancer. In subsequent studies, we will continue to expand the study population to further determine the prognostic role of LANR on patients with different stages of cervical cancer.

In summary, our study confirms that a high preoperative LANR may be an easily accessible new predictive factor for patients with stage IB-IIA cervical cancer. In clinical practice, LANR

may be considered a complementary indicator to the FIGO staging system and may help to identify high-risk patients in the same FIGO staged patients for early intervention to prolong the patient's survival cycle.

## Supporting information

**S1 Table. The clinical raw data of CC patients.**
(XLSX)

## Acknowledgments

The authors express gratitude to all study participants and research staff who participated in the work.

## Author Contributions

**Conceptualization:** Lei Wang.

**Data curation:** Shan Wang, Yuan Wang.

**Formal analysis:** Shan Wang, Yuan Wang, Jiaru Zhuang.

**Methodology:** Weifeng Shi.

**Project administration:** Weifeng Shi.

**Resources:** Yibo Wu.

**Supervision:** Lei Wang.

**Validation:** Yuan Wang, Jiaru Zhuang, Yibo Wu.

**Writing – original draft:** Shan Wang.

**Writing – review & editing:** Shan Wang.

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
