## [Decision Letter · Decision Letter 0]

27 Jul 2023

PONE-D-23-13687Prognostic Significance of Index (LANR) Composed of Preoperative Lymphocytes, Albumin, and Neutrophils in Patients With Stage IB-IIA Cervical CancerPLOS ONE

Dear Dr. Lei Wang,

Thank you for submitting your manuscript to PLOS ONE. After careful consideration, we feel that it has merit but does not fully meet PLOS ONE’s publication criteria as it currently stands. Therefore, we invite you to submit a revised version of the manuscript that addresses the points raised during the review process.

1. In the Introduction you should better explain the epidemiology of cervical cancer, otherwise is misleading. 2. The international ESGO guidelines suggests to not perform surgery in LACC. You should specify this.3. Why did you chose to include only FIGO IB and IIA patients? Explain it please4. FIGO IB3 is not include in the result table, please do it5. If there is a parametrial invasion the stage is IIB, did you exclude those patients after finding this out?6. Why did you perform laparoscopic surgery if LACC trial does not recommend this?7. It could be interesting if you do a comparison with other gynecologycal tumor.

We look forward to receiving your revised manuscript.

Kind regards,

Federico Romano, M.D., Ph.D.

Academic Editor

PLOS ONE

Journal Requirements:

This research was supported by a grant from the National Natural Science Fund (81701511) and grants from the Top Talent Support Program for Young and Middle-Aged People of Wuxi Health Committee (BJ2020047) and the Young Fund of Wuxi Health Commission (Q201926).

However, funding information should not appear in the Acknowledgments section or other areas of your manuscript. We will only publish funding information present in the Funding Statement section of the online submission form. 

This research was supported by a grant from the National Natural Science Fund (81701511) and grants from the Top Talent Support Program for Young and Middle-Aged People of Wuxi Health Committee (BJ2020047) .The funders had no role in study design, data collection and analysis, decision to publish, or preparation of the manuscript.

Reviewers' comments:

Reviewer's Responses to Questions

**Comments to the Author**

1. Is the manuscript technically sound, and do the data support the conclusions?

Reviewer #1: Yes

2. Has the statistical analysis been performed appropriately and rigorously? 

Reviewer #1: Yes

3. Have the authors made all data underlying the findings in their manuscript fully available?

Reviewer #1: Yes

4. Is the manuscript presented in an intelligible fashion and written in standard English?

Reviewer #1: No

5. Review Comments to the Author

Reviewer #1: In the present study, the authors retrospectively analyzed a cohort of 202 cervical cancer patients, and found that LANR is a independent prognostic factor for OS and PFS. This study is well designed and the result is of potential clinical significance. The questions need to be addressed are as follows:

1. The follow-up schedule should be prescribed in the follow-up section.

2. There are a lot of grammatical errors impairing understanding of the manuscript, and language editing is needed.

3. In fig. 2, it should be OS, not 0S.

6. PLOS authors have the option to publish the peer review history of their article (what does this mean?). If published, this will include your full peer review and any attached files.

Reviewer #1: **Yes: **Tao Hou

---

## [Author Response · Author response to Decision Letter 0]

3 Aug 2023

Academic editor:

1. In the Introduction you should better explain the epidemiology of cervical cancer, otherwise is misleading.

A: Thank you so much for your valuable advice. We have reviewed the literature carefully and added more relevant descriptions of the epidemiology of cervical cancer into the INTRODUCTION part of the revised manuscript.

Line 51-57, “Cervical cancer incidence and mortality rates vary significantly from country (or region) to country (or region) and are closely related to the level of socio-economic development, with more than 80 percent of new cases and deaths from cervical cancer occurring in low- and middle-income countries or regions. The incidence and mortality rates of cervical cancer increase as women age, with women aged 35-64 years being the main incidence group, and the incidence peaks at the age of 50-54 years.” was added.

1） Arbyn M, Weiderpass E, Bruni L, de Sanjosé S, Saraiya M, Ferlay J, et al. Estimates of incidence and mortality of cervical cancer in 2018: a worldwide analysis. Lancet Glob Health. 2020;8(2):e191-e203. 

2. The international ESGO guidelines suggests to not perform surgery in LACC. You should specify this.

A: We sincerely appreciate the significant suggestion. To answer the concern of the reviewer, we have made our conclusions more complete by adding a description of the LACC treatment in the DISCUSSION part.

Line 227-229, “The latest international ESGO guideline suggests not performing surgery in the locally advanced cervical cancer (LACC). Simultaneous radiotherapy and chemotherapy should be preferred for patients with narrowly conceptualized locally advanced (stage IB3 and IIA2) cervical cancer.” was added.

3. Why did you chose to include only FIGO IB and IIA patients? Explain it please

A： Thank you very much for your question. We aimed to study the prognostic effect of preoperative LANR on cervical cancer. Stage IA-IIA cervical cancer is an early-stage cervical cancer that can be cured with radical surgery or radiation. The surgical method of stage IA can be selected according to whether LVSI is accompanied and whether the patient has fertility requirements. Moreover, clinical studies have shown that the five-year survival rate of stage IA is higher, so it is not included. The standard treatment for stage IB1, IB2, and IIA1 is C-type radical hysterectomy and pelvic lymphadenectomy. The preferred treatment for stage IB3 and IIA2 is CCRT, but radical hysterectomy and pelvic lymphadenectomy can still be performed after NACT in areas lacking radiotherapy. To sum up, we chose patients with stage IB-IIA as the study object.

4. FIGO IB3 is not include in the result table, please do it

A: Thank you very much for your pertinent advice. In the section on Materials and Methods, we have indicated that the patients included in this study were based on the 2009 FIGO staging system, and stage IB was divided into IB1 and IB2 (corresponding to stage IB3 of the new 2018 FIGO staging system). As this is a retrospective study of medical records and archived samples, we did not restage patients.

5. If there is a parametrial invasion the stage is IIB, did you exclude those patients after finding this out?

A: Thank you very much for your question. We have not excluded this group of patients. Currently, clinical use of imaging tests such as CT, MRI, and PET for pre-treatment evaluation of cervical cancer has improved the accuracy of paracervical infiltration, but there are still some errors. In China's cervical cancer clinical diagnosis and treatment data, the rate of paracervical infiltration is about 1.9% in patients with stage IA1 to IIA2. In this paper, the rate of paracervical infiltration was 3.0% (stage IB1-IIA2), which we considered to be within the normal range and therefore retained these patients.

6. Why did you perform laparoscopic surgery if LACC trial does not recommend this?

A： Thank you very much for your question. China has a huge population base and the number of cervical cancer patients is about 1/3 of the world, but the LACC trial did not include clinical data from Chinese medical centers. Therefore a road-machine controlled study (NCT03739944) performed in China, which started collecting cases in December 2018, will compare the efficacy of minimally invasive surgery (including robotic surgery versus laparoscopic surgery) and open surgery for the treatment of early-stage cervical cancer, with strict criteria for participating surgeons, to overcome the influence of heterogeneity in surgeon experience on the study findings.

In addition, the Chinese Society of Gynaecological Oncology of the Chinese Medical Association released the Chinese Expert Consensus on Minimally Invasive Surgery for Cervical Cancer in 2019, which suggests that, on the one hand, attention should be paid to the conclusions of the LACC trial, and the indications for minimally invasive surgery versus open surgery for the treatment of cervical cancer should be rigorously grasped, while on the other hand, the value of minimally invasive surgery for the treatment of cervical cancer cannot be denied because of this, and the patient's prognosis should be improved by continually improving minimally invasive surgical methods.

In conclusion, until new high-level clinical evidence is available, we should choose our surgical procedures carefully to minimize and avoid the risks that minimally invasive surgery poses to both surgeons and patients.

2） Chao X, Li L, Wu M, Ma S, Tan X, Zhong S, et al. Efficacy of different surgical approaches in the clinical and survival outcomes of patients with early-stage cervical cancer: protocol of a phase III multicentre randomised controlled trial in China. BMJ Open. 2019;9(7):e029055.

3） Chinese expert consensus on minimally invasive surgery for cervical cancer [J]. Advances in Modern Obstetrics and Gynecology. 2019,28(11):801-803.

7. It could be interesting if you do a comparison with other gynecologycal tumor.

A： Your suggestion means a lot to us. Yes, it would be more interesting to make comparisons with other gynecological tumors and we will follow up the study afterwards.

Journal Requirements:

A: For requirements from the journal, we have ensured that the manuscript meets the style requirements of PLOS ONE.

2. Please provide additional details regarding participant consent. If you are reporting a retrospective study of medical records or archived samples, please ensure that you have discussed whether all data were fully anonymized before you accessed them and/or whether the IRB or ethics committee waived the requirement for informed consent.

A: What we report is a retrospective study of medical records and archived samples. It has been stated in the Materials And Methods（Line 90-93） of the manuscript that all data were anonymous before we accessed them, and the ethics committee waived the requirement of informed consent. The same is stated in the ethics statement labeled ' Medical Ethics Committee Notification' in the online submission.

3. Please remove any funding-related text from the manuscript and let us know how you would like to update your Funding Statement.

A: Thanks for your reminder, we have deleted the funding information in the manuscript, and the funding statement does not need to be updated.

Reviewer #1:

1. The follow-up schedule should be prescribed in the follow-up section.

A: We sincerely thank the reviewer for careful reading. Line 108-112, “The first post-surgical follow-up for cervical cancer was approximately two months after surgery, and patients were required to review pelvic CT, MRI, and blood tests including tumor markers, and subsequently every three months. If there was no recurrence within two years, it could be repeated every six months until five years. The follow-up period was defined as the date of treatment initiation to the date of final confirmation of patient survival or death.” was added.

2. There are a lot of grammatical errors impairing understanding of the manuscript, and language editing is needed.

A: Thanks for your suggestion. We feel sorry for our poor writings, however, we invited a college English teacher to help us polish our article. And we hope the revised manuscript could be acceptable for you.

3. In fig. 2, it should be OS, not 0S.

A: We feel sorry for our carelessness. In our resubmitted manuscript, the typo is revised. Thanks for your correction.

---

## [Editor Report · Decision Letter 1]

18 Aug 2023

Prognostic Significance of Index (LANR) Composed of Preoperative Lymphocytes, Albumin, and Neutrophils in Patients With Stage IB-IIA Cervical Cancer

PONE-D-23-13687R1

Dear Dr. Lei Wang,

We’re pleased to inform you that your manuscript has been judged scientifically suitable for publication and will be formally accepted for publication once it meets all outstanding technical requirements.

Kind regards,

Federico Romano, M.D., Ph.D.

Academic Editor

PLOS ONE

---

## [Editor Report · Acceptance letter]

11 Sep 2023

PONE-D-23-13687R1 

Prognostic Significance of Index (LANR) Composed of Preoperative Lymphocytes, Albumin, and Neutrophils in Patients With Stage IB-IIA Cervical Cancer 

Dear Dr. Wang:

I'm pleased to inform you that your manuscript has been deemed suitable for publication in PLOS ONE. Congratulations! Your manuscript is now with our production department. 

Kind regards, 

on behalf of

Dr. Federico Romano 

Academic Editor

PLOS ONE